

# Cooling phonon modes of a Bose condensate with uniform few body losses

Isabelle Bouchoule[1*], Max Schemmer[1] and Carsten Henkel[2]

**1** Laboratoire Charles Fabry, Institut d'Optique, CNRS, Université Paris Saclay,
2 Avenue Augustin Fresnel, 91127 Palaiseau Cedex, France
**2** Institute of Physics and Astronomy, University of Potsdam,
Karl-Liebknecht-Str. 24/25, 14476 Potsdam, Germany

⋆ isabelle.bouchoule@institutoptique.fr

## Abstract

We present a general analysis of the cooling produced by losses on condensates or quasi-condensates. We study how the occupations of the collective phonon modes evolve in time, assuming that the loss process is slow enough so that each mode adiabatically follows the decrease of the mean density. The theory is valid for any loss process whose rate is proportional to the $j$th power of the density, but otherwise spatially uniform. We cover both homogeneous gases and systems confined in a smooth potential. For a low-dimensional gas, we can take into account the modified equation of state due to the broadening of the cloud width along the tightly confined directions, which occurs for large interactions. We find that at large times, the temperature decreases proportionally to the energy scale $mc^2$, where $m$ is the mass of the particles and $c$ the sound velocity. We compute the asymptotic ratio of these two quantities for different limiting cases: a homogeneous gas in any dimension and a one-dimensional gas in a harmonic trap.


# 1 Introduction

Despite their extensive use as quantum simulators or for quantum sensing, the temperatures reached in ultracold gases are not fully understood. Careful analyses of the cooling mechanisms have a long tradition in the cold atoms community, and the corresponding temperature limits constitute important benchmarks. The role of atom losses, however, is not yet elucidated, although such processes often play a role in quantum gas experiments. Different loss processes may occur. One-body processes are always present, their origin could be for instance a collision with a hot atom from the residual vapour. The familiar method of evaporative cooling involves losses that depend on the particle energy, a case we exclude in this paper. For clouds trapped in an internal state which is not the lowest energy state, such as low-field seekers in a magnetic trap, two-body (spin flip) collisions may provide significant loss. Finally, three-body processes where atoms recombine into strongly bound dimers are always present and are often the dominant loss mechanism. The effect of one-body losses for an ideal Bose gas was investigated in [1]. Loss processes involving more than one body are a source of heating for trapped thermal clouds, since they remove preferentially atoms in dense regions where the potential energy is low [2]. Here we are interested in the effect of losses in Bose condensates or quasi-condensates, and we focus on low energy collective modes, whose physics is governed by interactions between atoms.

One-body losses have recently been investigated for one-dimensional (1D) quasi-condensates [3–6]. Quasi-condensates characterise weakly interacting 1D Bose gases at low enough temperature: repulsive interactions prevent large density fluctuations such that the gas resembles locally a Bose Einstein condensate (BEC), although it does not sustain true long-range order [7,8]. The above studies have focussed on low-energy excitations in the gas, the phonon modes. These correspond to hydrodynamic waves propagating in the condensate, where long-wavelength phase (or velocity) modulations are coupled to density modulations. On the one hand, losses reduce density fluctuations and thus remove interaction energy from each phonon mode. This decrease in energy, and thus of quasiparticle occupation, amounts to a cooling of the modes. On the other hand, the shot noise due to the discrete nature of losses feeds additional density fluctuations into the gas. This increases the energy per mode and amounts to heating. Theoretical studies [4–6], valid for one-body losses in 1D homogeneous gases, predict that as a net result of these competing processes, the system is cooling down in such a way that the ratio between temperature $k_B T$ and the chemical potential $\mu$ becomes asymptotically a constant (equal to 1). Many questions remain open. For instance, the role of longitudinal confinement has not been elucidated. Moreover, theoretical predictions for higher-body loss processes are lacking, although cooling by three-body losses was recently demonstrated experimentally [9].

In this paper, we generalise the theoretical results for one-body losses in homogeneous 1D gases and extend the analysis to a BEC or a quasicondensate in any dimension, for any

$j$-body loss process, and for homogeneous gases as well as clouds confined in a smoothly varying trapping potential. We concentrate on phonon modes and the loss rate is assumed small enough to ensure adiabatic following of each mode. Low-dimensional systems are realised experimentally by freezing the transverse degrees of freedom with a strong transverse confinement. However, in many experiments the interaction energy is not negligible compared to the transverse excitation frequencies such that the freezing is not perfect. The interactions then broaden the wave function in the transverse directions, and phonon modes in the weakly confined directions are associated with transverse breathing [10–12]. Our theory can take this into account with a modified equation of state: the quantities $\mu$ and $mc^2$, where $m$ is the atomic mass and $c$ the sound velocity, equal for a strong transverse confinement, no longer coincide. We find that the evolution produced by losses is better described by a constant ratio $k_B T/(mc^2)$ instead of $k_B T/\mu$. The asymptotic ratio $k_B T/(mc^2)$ is computed for a few examples. Predictions from this paper have been tested successfully against recent experimental results obtained at Laboratoire Charles Fabry on the effect of three-body losses in a harmonically confined 1D Bose gas [9].

## 2 Model

We consider a condensate, or quasi-condensate, in dimension $d = 1, 2$ or $3$. The gas is either homogeneous or trapped in a smoothly varying potential $V(\mathbf{r})$. We assume it is subject to a $j$-body loss process of rate constant $\kappa_j$: the number of atoms lost per unit time and unit volume is $\kappa_j n^j$ where $n$ is the density. This density includes fluctuations of quantum and thermal nature, and its average profile is denoted $n_0(\mathbf{r}, t)$. Instead of using involved powerful theoretical techniques such as the truncated Wigner approach [13,14], we compute the effect of losses in this paper with a spatially coarse-grained approach that does not rely on involved theory and in which the approximations are made transparent. For the same pedagogical reason, we explicitly construct the phase-density representation of the collective excitations of the gas, in a similar way as is done for instance in [15].

### 2.1 Stochastic dynamics of the particle density

Let us first consider the sole effect of losses and fix a cell of the gas of volume $\Delta$, small enough so that the density of the (quasi)condensate is about homogeneous in this volume, but large enough to accommodate many atoms. The atom number in the cell is $N = N_0 + \delta N$ where $N_0 = n_0 \Delta$ and $\delta N \ll N_0$ since the gas lies in the (quasi)condensate regime. (We drop the position dependence $n_0 = n_0(\mathbf{r})$ for the moment.) Since typical values of $\delta N$ are much smaller than $N_0$, one can assume without consequence that $\delta N$ is a variable that takes discrete values between $-\infty$ and $\infty$. Hence, one can define a phase operator $\theta$, whose eigenvalues span the interval $[0, 2\pi[$ and that is canonically conjugate to $\delta N$. Losses will affect both the density fluctuations and the phase fluctuations.

We first concentrate on the effect of losses on density fluctuations. Consider a time step $dt$, small enough that the change $dN$ in atom number is much smaller than $N$, but large enough such that $dN$ is much larger than 1. After the time step, we have

$$dN = -K_j N^j dt + d\xi, \tag{1}$$

where $K_j = \kappa_j/\Delta^{j-1}$. Here, $d\xi$ is a random number with vanishing mean value that translates the shot noise associated with the statistical nature of losses. The number of loss events during the small step $dt$ is Poisson distributed so that the variance of $d\xi$ relates to the mean number of lost atoms by

$$\langle d\xi^2 \rangle = jK_j N^j dt \simeq jK_j N_0^j dt, \tag{2}$$

the factor $j$ coming from the fact that at each event, $j$ atoms are lost. The evolution of fluctuations in the atom number is obtained from $d\delta N = dN - dN_0$, where $dN_0$ is the change of the mean number, equal to $dN_0 = -K_j N_0^j dt$ in the lowest order in $\delta N$. Expanding $N^j$ in Eq.(1) to first order in $\delta N$, we obtain the following evolution for the density fluctuation $\delta n = \delta N/\Delta$:

$$d\delta n = -j\kappa_j n_0^{j-1}\delta n\, dt + d\eta, \tag{3}$$

where $d\eta = d\xi/\Delta$ is a random variable of variance $\langle d\eta^2\rangle = j\kappa_j n_0^j dt/\Delta$. The first term in the r.h.s, the drift term, decreases the density fluctuations. It will thus reduce the interaction energy associated to fluctuations in the gas and produce cooling. The second term on the other hand increases the density fluctuations in the gas which leads to heating.

## 2.2 Shot noise and phase broadening

We now compute the effect of losses on the phase fluctuations, following an approach similar to Ref. [16]. For this purpose, one imagines that one records the number of lost atoms during $dt$. This measurement increases the knowledge about $N$, and thus $\delta N$. To quantify this increase of knowledge, we use the Bayes formula

$$P(\delta N|N_l) = \frac{P(\delta N)}{\int d(\delta N')P(N_l|\delta N')}P(N_l|\delta N), \tag{4}$$

where $P(\delta N)$ is the initial probability of having an atom number $N = N_0 + \delta N$, and $P(N_l|\delta N)$ is the probability that a number $N_l$ of atoms will be lost, given that the initial atom number was $N_0 + \delta N$. Finally, $P(\delta N|N_l)$ is the probability that the final number is $N_0 - N_l + \delta N$, knowing the fact that $N_l$ atom have been lost. As argued above, the Poissonian nature of the loss process and the assumption that the number of lost atoms is large compare to one, imply the Gaussian distribution

$$P(N_l|\delta N) \simeq \frac{1}{\sqrt{2\pi}\sigma_l}e^{-(N_l-K_j N^j dt)^2/(2\sigma_l^2)}, \tag{5}$$

where $N = N_0 + \delta N$ and $\sigma_l^2 = jK_j N_0^j dt$. Expanding $N^j$ around $N_0^j$ and introducing $\overline{\delta N} = N_l/(jK_j N_0^{j-1}dt) - N_0/j$, one has

$$\frac{(N_l - K_j N^j dt)^2}{\sigma_l^2} \simeq \frac{(\overline{\delta N} - \delta N)^2}{\sigma_{\delta N}^2}, \tag{6}$$

where

$$\sigma_{\delta N}^2 = \frac{N_0}{jK_j N_0^{j-1}dt}. \tag{7}$$

Thus, according to Eq.(4), the width of the distribution in $\delta N$ is multiplied by a function of rms width $\sigma_{\delta N}$ after recording the number of lost atoms. This narrows the number distribution and must be associated with a broadening in the conjugate variable, $\theta$, lest the uncertainty relations are violated. The phase broadening must be equal to

$$\langle d\theta^2\rangle = \frac{1}{4\sigma_{\delta N}^2} = \frac{j\kappa_j n_0^{j-1}}{4n_0\Delta}dt. \tag{8}$$

This spreading of the phase results from the shot noise in the loss process.

In the following, keeping in mind that only length scales larger than the interparticle distance have to be considered, we go to the continuous limit. The factors $1/\Delta$ in the variance

for $d\eta$ in Eq.(3) and in the phase diffusion of Eq.(8) then turn into

$$\langle d\eta(\mathbf{r})d\eta(\mathbf{r}')\rangle = j\kappa_j n_0^j \delta(\mathbf{r}-\mathbf{r}')dt, \tag{9}$$

$$\langle d\theta(\mathbf{r})d\theta(\mathbf{r}')\rangle = \frac{j}{4}\kappa_j n_0^{j-2}\delta(\mathbf{r}-\mathbf{r}')dt. \tag{10}$$

Both diffusion terms are due to the quantised nature of the bosonic field, namely the discreteness of atoms. Their effects become negligible compared to the drift term in Eq. (3) in the classical field limit, i.e. $n_0 \to \infty$ at fixed typical density fluctuations $\delta n/n_0$. Note finally that these results could also have been obtained using a truncated Wigner approach [13,14], using approximations based on the relation $\delta n \ll n_0$.

Before going on, let us make a remark concerning gases in reduced dimension. An effective 1D (resp. 2D) gas is obtained using a strong transverse confinement in order to freeze the transverse degree of freedom: the atoms are in the transverse ground state of the confining potential, of wave function $\psi(x_\perp)$. In the case of $j$-body losses with $j > 1$, the loss process a priori modifies the transverse shape of the cloud since it occurs preferentially at the center, where the density is the highest. In other words, it introduces couplings towards transverse excitations. We assume here the loss rate to be much smaller than the frequency gap $\omega_\perp$ between the transverse ground and first excited states. Then the coupling to transverse excitations has negligible effects, and the above analysis of the effect of losses also holds for the effective 1D (resp. 2D) gas, provided $\kappa_j = \kappa_j^{3D}\int d^2x_\perp|\psi(x_\perp)|^{2j}$ (resp. $\kappa_j = \kappa_j^{3D}\int dx_\perp|\psi(x_\perp)|^{2j}$), where $\kappa_j^{3D}$ is the rate constant coefficient for the 3D gas.

## 2.3 Collective excitations

Let us now take into account the dynamics of the gas. Under the effect of losses the profile $n_0(\mathbf{r},t)$ evolves in time and, except for a homogeneous system, a mean velocity field appears, generated by a spatially dependent phase $\theta_0(\mathbf{r},t)$. Here we assume the loss rate is small enough so that, at any time, $n_0(\mathbf{r})$ is close to the equilibrium profile. We moreover assume the potential varies sufficiently smoothly such that the equilibrium profile is obtained with the local density approximation. Then, at any time, $n_0(\mathbf{r})$ fulfills

$$\mu(n_0(\mathbf{r})) = \mu_p - V(\mathbf{r}), \tag{11}$$

where $\mu(n)$ is the chemical potential of a homogeneous gas of density $n$ and $\mu_p$ is the peak chemical potential, which fixes the total atom number [1]. In most cases $\mu = gn$ where $g$ is the coupling constant. In 3D condensates, $g = 4\pi\hbar^2 a/m$ where $a$ is the scattering length describing low-energy collisions. In situations where two (resp. one) degrees of freedom are strongly confined by a transverse potential of frequency $\omega_\perp$, $\mu$ depends on $a$, on the linear (resp. surface) density $n$, and on $\omega_\perp$. As long as $\hbar\omega_\perp \gg \mu$, the transverse cloud shape is close to that of the transverse ground state [2], and one recovers the expression $\mu = gn$ where the effective 1D (resp. 2D) coupling constant $g$ depends only on $a$ and on $\omega_\perp$ [17,18]. At large densities, $\hbar\omega_\perp \sim \mu$, the transverse degrees of freedom are no longer completely frozen: interactions broaden the transverse wave function, and $\mu$ is no longer linear in $n$ [11,12]. We discuss one example in Sec.3.2.

To treat the dynamics around the average density $n_0(\mathbf{r},t)$, a Bogoliubov approximation is valid since the gas is in the (quasi)condensate regime: one can linearise the equations of motion in the density and phase fluctuations $\delta n(\mathbf{r})$ and $\varphi(\mathbf{r}) = \theta - \theta_0$ [15,19]. These equations

---

[1]The peak density is reached at the position $\mathbf{r}_p$ where $V$ reaches its minimum value. We impose $V(\mathbf{r}_p) = 0$.

[2]We assume here that the transverse width of the cloud fulfills $l_\perp \gg a$ such that the effect of interactions is well captured treating the gas as a 3D gas.

involve the mean velocity field $\hbar\nabla\theta_0/m$. Here we assume the loss rate is small enough so that such terms are negligible. We moreover consider only length scales much larger than the healing length. Then, as detailed in Appendix A, the dynamics of $\delta n(\mathbf{r})$ and $\varphi(\mathbf{r})$ is governed by the hydrodynamic Hamiltonian

$$H_{\text{hdyn}} = \frac{\hbar^2}{2m}\int d^d\mathbf{r}\, n_0\,(\nabla\varphi)^2 + \frac{m}{2}\int d^d\mathbf{r}\frac{c^2}{n_0}\delta n^2. \tag{12}$$

Here the speed of sound $c = c(\mathbf{r})$ is related to the local compressibility, $mc^2 = n_0\partial_n\mu$, evaluated at $n_0(\mathbf{r})$. At a given time, $H_{\text{hdyn}}$ can be recast as a collection of independent collective modes. The collective modes are described by the eigenfrequencies $\omega_\nu$ and the real functions $g_\nu$ [details in Appendix B]. They obey

$$\nabla\cdot\left(n_0\nabla(\frac{c^2}{n_0}g_\nu)\right) = -\omega_\nu^2 g_\nu, \tag{13}$$

and are normalised according to

$$\delta_{\nu,\nu'} = \frac{m}{\hbar\omega_\nu}\int d^d\mathbf{r}\frac{c^2}{n_0}g_\nu(\mathbf{r})g_{\nu'}(\mathbf{r}). \tag{14}$$

Then $H_{\text{hdyn}} = \sum_\nu H_\nu$ where

$$H_\nu = \frac{\hbar\omega_\nu}{2}(x_\nu^2 + p_\nu^2). \tag{15}$$

The dimensionless canonically conjugate quadratures $x_\nu$ and $p_\nu$ are related to $\delta n$ and $\varphi$ respectively. More precisely,

$$\begin{cases} \delta n(\mathbf{r}) = \sum_\nu x_\nu g_\nu(\mathbf{r}) \\ \varphi(\mathbf{r}) = \frac{mc^2}{n_0}\sum_\nu p_\nu\frac{g_\nu(\mathbf{r})}{\hbar\omega_\nu}, \end{cases} \tag{16}$$

which inverts into

$$\begin{cases} x_\nu = \frac{m}{\hbar\omega_\nu}\int d^d\mathbf{r}\frac{c^2}{n_0}\delta n(\mathbf{r})g_\nu(\mathbf{r}) \\ p_\nu = \int d^d\mathbf{r}\,\varphi(\mathbf{r})g_\nu(\mathbf{r}). \end{cases} \tag{17}$$

At thermal equilibrium, the energy in the mode $\nu$ is equally shared between both quadratures and, for temperatures $T \gg \hbar\omega_\nu$, one has $\langle H_\nu\rangle = T$.

# 3 Cooling dynamics

## 3.1 Evolution of the excitations

Let us consider the effect of losses on the collective modes. The loss process modifies in time the mean density profile and thus the two functions of $\mathbf{r}$, $n_0$ and $c$, that enter into the Hamiltonian Eq. (12). We however assume the loss rate is very low compared to the mode frequency and their differences $\omega_\nu - \omega_{\nu'}$, so that the system follows adiabatically the effect of these modifications. As a consequence, equipartition of the energy holds at all times for any collective mode $\nu$, and the adiabatic invariant $A_\nu = \langle H_\nu\rangle/(\hbar\omega_\nu)$ is unaffected by the slow evolution of $n_0$. The dynamics of $A_\nu$ is then only due to the modifications of $\delta n(\mathbf{r})$ and $\varphi(\mathbf{r})$ induced by the loss process (subscript $l$), namely

$$\frac{dA_\nu}{dt} = \frac{1}{2}\left(\frac{d\langle x_\nu^2\rangle_l}{dt} + \frac{d\langle p_\nu^2\rangle_l}{dt}\right). \tag{18}$$

Injecting Eq. (3) into Eq. (17), we obtain for the 'density quadrature'

$$(dx_\nu)_l = \frac{m}{\hbar\omega_\nu} \int d^d\mathbf{r} \frac{c^2}{n_0} g_\nu(\mathbf{r}) \left(-j\kappa_j n_0^{j-1} \delta n(\mathbf{r}) dt + d\eta(\mathbf{r})\right). \tag{19}$$

Using the mode expansion (16) for $\delta n(\mathbf{r})$ in the first term, we observe the appearance of couplings between modes. In the adiabatic limit (loss rate small compared to mode spacing), the effect of these couplings is however negligible. Then, Eq. (19) leads to

$$\frac{d\langle x_\nu^2\rangle_l}{dt} = -\frac{2j\kappa_j m}{\hbar\omega_\nu}\langle x_\nu^2\rangle \int d^d\mathbf{r} c^2 n_0^{j-2} g_\nu^2 + \frac{j\kappa_j m^2}{(\hbar\omega_\nu)^2} \int d^d\mathbf{r} c^4 n_0^{j-2} g_\nu^2. \tag{20}$$

Let us now turn to the phase diffusion associated with losses. It modifies the width of the conjugate quadrature $p_\nu$, according to

$$\frac{d\langle p_\nu^2\rangle_l}{dt} = \frac{j\kappa_j}{4} \int d^d\mathbf{r}\, n_0^{j-2} g_\nu^2. \tag{21}$$

The hydrodynamic modes are characterised by low energies, $\hbar\omega_\nu \ll mc^2$, when the speed of sound is evaluated in the bulk of the (quasi)condensate. Then $d\langle p_\nu^2\rangle_l/dt$ gives a contribution that scales with the small factor $(\hbar\omega_\nu/mc^2)^2$ compared to the second term of Eq. (20). In other words one expects that the phase diffusion associated to the loss process gives a negligible contribution to the evolution of $A_\nu$ [Eq.(18)] for phonon modes [3].

We see from Eq.(20) that the adiabatic invariant $A_\nu$ is actually changed by $j$-body losses. We now show that the decrease in the energy per mode $\langle H_\nu\rangle$ is better captured by the energy scale associated with the speed of sound, as their ratio will converge towards a constant during the loss process. More precisely, we introduce

$$y_\nu = \frac{\langle H_\nu\rangle}{mc_p^2} \simeq \frac{k_B T_\nu}{mc_p^2}, \tag{22}$$

where $c_p$ is the speed of sound evaluated at the peak density $n_p$. The second expression is valid as long as the phonon modes stay in the classical regime, $\langle H_\nu\rangle \gg \hbar\omega_\nu$. From Eq. (18) and (20), neglecting the contribution of Eq.(21), we immediately obtain

$$\frac{d}{dt} y_\nu = \kappa_j n_p^{j-1} \left[ -(j\mathscr{A} - \mathscr{C})y_\nu + j\mathscr{B} \right], \tag{23}$$

where the dimensionless parameters $\mathscr{A}, \mathscr{B}$ and $\mathscr{C}$ are

$$\mathscr{A} = \frac{m}{\hbar\omega_\nu} \int d^d\mathbf{r} \frac{c^2 n_0^{j-2}}{n_p^{j-1}} g_\nu^2(\mathbf{r}), \tag{24}$$

$$\mathscr{B} = \frac{m}{2\hbar\omega_\nu} \int d^d\mathbf{r} \frac{c^4 n_0^{j-2}}{c_p^2 n_p^{j-1}} g_\nu^2(\mathbf{r}), \tag{25}$$

$$\mathscr{C} = \frac{d\ln(mc_p^2/\hbar\omega_\nu)}{dN_{\text{tot}}} \int d^d\mathbf{r} \frac{n_0^j}{n_p^{j-1}}. \tag{26}$$

In general, all of them depend on $\nu$ but we omit the index $\nu$ for compactness. The term $\mathscr{A}$ is the rate of decrease of $y_\nu$ induced by the reduction of the density fluctuations under the loss

---

[3]At the border of the (quasi)condensate, where the density becomes small, the condition $\hbar\omega_\nu \ll mc^2$ breaks down, however. The effect of phase diffusion is more carefully evaluated in Sec.3.3.

process, normalised to $\kappa_j n_p^{j-1}$. The term $\mathscr{B}$ originates from the additional density fluctuations induced by the stochastic nature of the losses. The term $\mathscr{C}$ arises from the time dependence of the ratio $mc_p^2/\hbar\omega_\nu$. It is computed using the dependence of $mc_p^2/\hbar\omega_\nu$ on the total atom number, the latter evolving according to

$$\frac{dN_{\text{tot}}}{dt} = -\int d^d\mathbf{r}\,\kappa_j n_0^j. \tag{27}$$

Eqs. (23–26) constitute the main results of this paper. They have been solved numerically for the experimental parameters corresponding to the data of [9] ($j=3$ and anisotropic harmonic confinement) and their predictions compare very well with experimental results.

We would like at this stage to make a few comments about these equations. First, the factor $\hbar$, although it appears explicitly in the equations, is not relevant since it is canceled by the $\hbar$ contained in the normalisation (14) of the mode functions $g_\nu$. Second, we note that $\mathscr{A}$, $\mathscr{B}$ and $\mathscr{C}$ are intensive parameters: they are invariant by a scaling transformation $V(\mathbf{r}) \to V(\lambda\mathbf{r})$ and depend only on the peak density $n_p$ and on the shape of the potential. Finally, Eqs. (23–26) depend on $\nu$ and it is possible that the lossy (quasi-)condensate evolves into a non-thermal state where different modes acquire different temperatures. Such a non-thermal state of the gas is permitted within the linearised approach where modes are decoupled. In the examples studied below, however, it turns out that all hydrodynamic modes share about the same temperature[4]. In the following, we investigate the consequences of Eq. (23-26), considering different situations.

## 3.2 Example: homogeneous gas

In this case, density $n_0$ and speed of sound $c$ are spatially constant. The collective modes are sinusoidal functions, labelled by $\nu$ and of wave vector $\mathbf{k}_\nu$ [5]. The frequencies are given by the acoustic dispersion relation $\omega_\nu = c|\mathbf{k}_\nu|$ and the mode functions $g_{\nu c,s}(\mathbf{r})$ are normalised to

$$\int d^d\mathbf{r}\,g_\nu^2(\mathbf{r}) = \frac{\hbar\omega_\nu}{mc^2}n_0. \tag{28}$$

Then Eqs.(23-26) reduce to

$$\frac{d}{dt}y = \kappa_j n_0^{j-1}\left[-y\left(j - \frac{\partial\log c}{\partial\log n_0}\right) + j/2\right], \tag{29}$$

which is the same for all modes $\nu$. Let us consider the limit $\mu = gn_0$, valid in 3D gases, or in low-dimensional gases with strong transverse confinement (negligible broadening of the transverse wave function). Then $c \propto n_0^{1/2}$ and Eq. (29) shows that $y$ tends at long times towards the asymptotic value

$$y_\infty = \frac{1}{2 - 1/j}, \tag{30}$$

independent of the mode energy[4]. For one-body losses, one recovers the result $y_\infty = 1$ [4,5]. In the case of 3-body losses, one finds $y_\infty = 3/5$.

Let us now consider a quasi-low-dimensional gas, where transverse broadening of the wave function cannot be neglected. The logarithmic derivative in Eq.(29) is then no longer constant.

---

[4] In the case of one-body losses, theories that go beyond the hydrodynamic approximation predict non-thermal states to appear, where the high-frequency modes reach higher temperatures than the phonon modes [4,5].

[5] For 1D gases, $\nu = (p,\sigma)$ where $p$ is a positive integer and $\sigma = c$ or $s$ depending wether we consider cosine or sine modes. The wave-vector is $k_\nu = 2p\pi/L$ where $L$ is the length of the box, assuming periodic boundary conditions. This generalises to higher dimensions with $\nu = (p_1,\sigma_1,p_2,\sigma_2,p_3,\sigma_3)$ in 3D for instance.

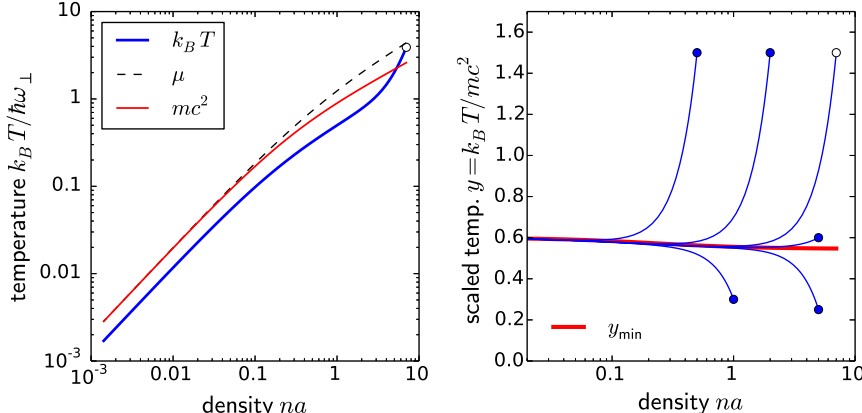

Figure 1: Cooling a quasi-1D gas, homogeneous along the axial direction, by three-body losses. The density is initially so high that transverse broadening is relevant [the chemical potential does not fulfill $\mu \ll \hbar\omega_\perp$]. Left: time evolution of the temperature (thick blue), shown versus the time decreasing density. Black dashed and thin red lines show the intrinsic energy scales $\mu$ and $mc^2$. The system rapidly evolves into a dynamical state where the temperature follows the energy scale $mc^2$, rather than the chemical potential. Right: evolution of the ratio $y = k_B T/mc^2$ vs. the density. The curves correspond to different initial values (marked with dots, the white dot corresponding to the parameters on the left). The thick red line shows the function $y_{\min}$ that gives the positions of lowest values taken by $y$ in the course of cooling. In this system (homogeneous along the axial direction), all hydrodynamic modes evolve with the same temperature.

We will focus on the case of a quasi-1D gas, as realised experimentally for instance in [9]. The effect of the transverse broadening is well captured by the heuristic equation of state [11,12]

$$\mu = \hbar\omega_\perp \left( \sqrt{1 + 4n_0 a} - 1 \right), \tag{31}$$

where $\omega_\perp$ is the frequency of the transverse confinement and $a$ the 3D scattering length. Inserting into Eq. (29), one can compute the evolution of $y$. The transverse broadening also modifies the rate coefficient $\kappa_j$, making it density-dependent. However, re-scaling the time according to $u = \int_0^t \kappa_j(\tau) n_p^{j-1} d\tau = \ln(n_0(0)/n_0(t))$, Eq. (29) transforms into

$$\frac{dy}{du} = -y \left( j - 1/2 + \frac{n_0(0) a\, e^{-u}}{1 + 4n_0(0) a\, e^{-u}} \right) + j/2 \tag{32}$$

and no longer depends on $\kappa_j$. Fig.1 shows the solution of this differential equation in the case of 3-body losses, and for a few initial situations, namely different values of $y$ and $n_0 a$ (right plot). The asymptotic value $y = y_\infty$ is always reached at long times since the transverse broadening then becomes negligible. Note that in distinction to pure 1D gases, the effect of transverse broadening allows the system to reach transiently lower scaled temperatures $y < y_\infty$, even when starting at values of $y$ larger than $y_\infty$. More precisely, let us denote $y_{\min}(n_0) = j/2/(j - 1/2 + an_0/(1 + 4an_0))$. When starting with $y > y_{\min}$, the lowest value of $y$ is reached for some (non-vanishing) density, and it falls on the curve $y_{\min}$. For $j = 3$, one find that $y_{\min}$ varies between $y_\infty = 0.6$ and $6/11 \simeq 0.55$. Thus, the coldest temperatures in the course of the loss process never deviate by more than 10% from the asymptotic value 0.6: the impact of transverse swelling is relatively small. Note that, if one considered the scaled temperature $T/\mu$ rather than $y$, much larger deviations would appear.

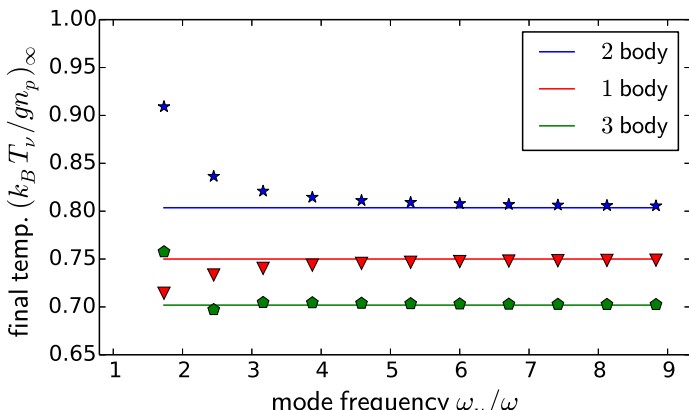

Figure 2: Asymptotic ratio $y_\infty = k_B T/mc^2$ for hydrodynamic collective modes of a 1D quasi-condensate confined in a harmonic trap, for 1-body (red), 2-body (blue) and 3-body (green) losses. The modes are labeled by their eigenfrequencies $\omega_\nu = \omega\sqrt{\nu(\nu+1)/2}$ and we only consider $\nu \geq 2$. Symbols: calculation based on the Legendre polynomials of Eq.(34), inserted into Eqs. (24, 25). Solid lines: large-$\nu$ approximation given by Eq. (36) with values $y_\infty = 3/4, 45/56, 525/748$ for $j = 1, 2, 3$.

## 3.3 Example: 1D harmonic trap

We consider a 1D gas confined in a harmonic potential of trapping frequency $\omega$. We assume for simplicity a pure 1D situation with $\mu = gn = mc^2$. In the Thomas-Fermi approximation, the mean density profile is

$$n_0(z) = n_p(1-(z/R)^2), \quad |z| \leq R, \tag{33}$$

where $n_p$ is the peak density and $R = \sqrt{2gn_p/(m\omega^2)}$ is the axial radius of the quasicondensate. From Eq.(13), we recover the known result that the hydrodynamic modes are described by the Legendre polynomials $P_\nu$, and the eigenfrequencies are $\omega_\nu = \omega\sqrt{\nu(\nu+1)/2}$ [7,20]. A trivial calculation using $N_{tot} = \frac{4}{3}n_p R \propto c_p^3$ and the substitution $z = R\cos\alpha$ gives $\mathscr{C} = \int_0^{\pi/2} d\alpha \sin^{2j+1}\alpha = 2/3, 8/15, 16/35$ for $j = 1, 2, 3$. To compute $\mathscr{A}$ and $\mathscr{B}$, one needs the exact expression of $g_\nu$, which according to the normalisation (14) can be written

$$g_\nu(z) = \sqrt{\frac{\hbar\omega_\nu}{2gR}}\sqrt{2\nu+1}P_\nu(z/R). \tag{34}$$

Inserting this expression, together with Eq. (33), into the integrals (24) and (25), we find that $\mathscr{A}$, $\mathscr{B}$, and $\mathscr{C}$ are time-independent. Thus $y$ tends at long times towards the asymptotic value $y_\infty = j\mathscr{B}/(j\mathscr{A} - \mathscr{C})$. For large $\nu$, one can use the asymptotic expansion [21]

$$P_\nu(\cos\alpha) \simeq \left(\frac{2}{\pi(\nu+\frac{1}{2})\sin\alpha}\right)^{1/2}\cos\phi_\nu, \tag{35}$$

with $\phi_\nu = (\nu+\frac{1}{2})\alpha - \frac{1}{4}\pi$. Moreover the fast oscillations of $P_\nu(\cos\alpha)$ can be averaged out in the calculation of the coefficients $\mathscr{A}$ and $\mathscr{B}$. Then $\mathscr{A}$ and $\mathscr{B}$ no longer depend on $\nu$, so that $y_\infty$ is identical for all modes, and we find

$$y_\infty \simeq \frac{\frac{j}{\pi}\int_0^{\pi/2}d\alpha\,\sin^{2j}\alpha}{\frac{2j}{\pi}\int_0^{\pi/2}d\alpha\,\sin^{2j-2}\alpha - \int_0^{\pi/2}d\alpha\,\sin^{2j+1}\alpha}. \tag{36}$$

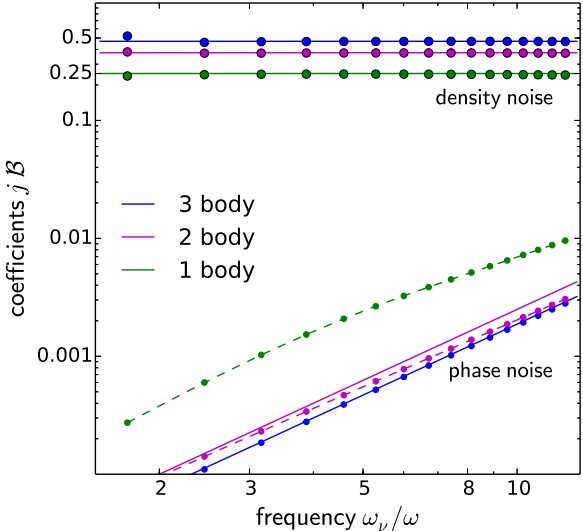

Figure 3: Diffusion of density and phase quadratures associated with many-body loss in a one-dimensional gas trapped in a harmonic potential. We plot the dimensionless coefficients $j\mathcal{B}$ [Eq.(25)] and $j\mathcal{B}_\varphi$ [Eq.(37)] that are proportional to the shot noise projected onto the corresponding quadratures. Symbols: numerically computed mode functions, improving upon the hydrodynamic approximation. Solid lines: approximate results based on the Legendre modes (34). Dashed lines: guide to the eye. Parameters: strictly 1D equation of state $\mu = gn$, peak chemical potential $\mu_p \approx gn_p = 100\hbar\omega$.

For one- and three-body losses, this gives $y_\infty = 3/4 = 0.75$ and $y_\infty = 525/748 \simeq 0.701$, respectively. This asymptotic result is compared to calculations using the expression Eq. (34) in Fig. 2. We find very good agreement as soon as the mode index is larger than 5.

To conclude this example, we come back to the diffusive dynamics of the 'phase quadratures' $p_\nu$ we neglected so far. In the case of one-body losses, however, it happens that the integral (21) does not converge: while the mode function $g_\nu(z)$ [Eq.(34)] remains finite at the condensate border $z \to \pm R$, the integrand $n_0^{j-2}(z)g_\nu^2(z)$ is not integrable for $j = 1$. This is actually an artefact of the hydrodynamic approximation, which breaks down at the border of the condensate.

We have performed numerical calculations of the collective excitations by solving the Bogoliubov equations [6]. The mode functions $g_\nu(z)$ are defined according to Eq.(61): they extend smoothly beyond the Thomas-Fermi radius and match well with the Legendre polynomials (34) within the bulk of the gas. The resulting values for the parameter $\mathcal{B}$ [Eq.(25)] are shown in Fig.3: they depend very weakly on the mode index $\nu$ and are well described by the approximate calculation based on the Legendre modes mentioned after Eq.(35) (solid lines). In the lower part of the figure, the corresponding values for the diffusion coefficient originating from phase noise are shown, namely the parameter

$$\mathcal{B}_\varphi = \frac{\hbar\omega_\nu}{8mc_p^2} \int d^d\mathbf{r} \frac{n_0^{j-2}}{n_p^{j-1}} g_\nu^2(\mathbf{r}). \tag{37}$$

They remain at least one order of magnitude below. For losses involving more than one particle, the approximation, under which the functions $g_\nu$ are given by the Legendre polynomials,

---

[6]For the condensate wave-function, we also went beyond the Thomas-Fermi approximation by allowing for a 'spill-over' of the condensate density beyond the inverted parabola.

gives a convergent integral in Eq.(37). The result is shown as solid lines for two- and three-body losses, where we made the additional approximation Eq. (35) on the Legendre functions and we averaged out the oscillating part. We find that the Legendre approximation performs better for three-body losses than for 2-body losses, which is expected since a stronger weight is given to the bulk rather than the edge of the condensate. In conclusion of this numerical study, we verified the validity of the assumption that, for phonon modes, the phase diffusion term gives negligible contribution to the evolution of $y$. This term becomes noticeable when one leaves the phonon regime $\hbar \omega_\nu \ll mc^2$. Then, one should go beyond the hydrodynamic Hamiltonian Eq.(12) to properly compute the mode dynamics.[7]

# 4 Conclusion

In this paper, we construct a stochastic model to describe the effect of losses on the hydrodynamic collective modes of condensates or quasicondensates. Explicit formulas for cooling and diffusion of the density and phase quadratures are derived. They provide the behaviour of the mode temperature $T$ with time. We show that $T$ becomes proportional to the energy scale $mc^2$ where $c$ is the hydrodynamic speed of sound. The asymptotic ratio $k_B T/(mc^2)$ is computed explicitly in different situations and for different $j$-body processes. These results are in good agreement with recent experiments performed in Laboratoire Charles Fabry [9] where three-body losses provided the dominant loss channel.

This work raises many different questions and remarks. First, it is instructive to investigate the evolution of the ratio $D = \hbar^2 n^{2/d}/(mk_B T)$, where $d$ is the gas dimension, since $D$ quantifies the quantum degeneracy of the gas.[8] Let us focus for simplicity on a homogeneous system and use $mc^2 = gn$. Once the ratio $k_B T/(mc^2)$ has become stationary, we find that $D$ increases in time for 3-dimensional gases, while it decreases for one-dimensional gases. Starting with a 1D Bose gas in the quasi-condensate regime, losses let the quantity $D\gamma$ reach a stationary value of order one, but increase the dimensionless interaction parameter $\gamma = mg/(\hbar^2 n)$. When $\gamma$, from values much smaller, approaches 1, the gas lies at the crossover between four regimes: the quasi-condensate ($\gamma \ll 1$, $D\sqrt{\gamma} \gg 1$), the quantum-degenerate ideal Bose gas ($D\sqrt{\gamma} \ll 1$, $D \gg 1$), the non-degenerate ideal Bose gas ($D\gamma^2 \ll 1$, $D \ll 1$) and the Tonks-Girardeau regime ($\gamma \gg 1$, $D\gamma^2 \gg 1$). At later times, one expects the cloud to leave the quasi-condensate regime and we believe it becomes a non-degenerate ideal Bose gas. Second, the effect of losses on high-frequency modes, not described by our hydrodynamic model, leads a priori to higher temperatures; this was investigated for 1D gases subject to one-body losses [5]. The gas is then described by a generalised Gibbs ensemble where different collective modes experience different temperatures. This non-thermal state is even long lived in 1D quasicondensates [5]. While the calculations presented here are formally valid for higher dimensions, efficient coupling between modes may reduce their relevance, since such coupling favours a common temperature. It is an open question whether our methods could be extended to the case of evaporative cooling where the one-body loss rate is energy- or position-dependent. This mechanism may play a role in experiments where temperatures as low as $k_B T \approx 0.3\, mc^2$ have been observed, lower than the predicted temperatures for uniform losses [3]. Finally, it would be interesting to extend this work to different regimes of the gas. For instance, one may ask how the effect of losses transforms as one goes from a quasi-condensate to the ideal gas regime. The approximation of weak density fluctuations then clearly becomes invalid. One could also investigate losses at even lower densities, where the 1D gas enters the fermionised (or Tonks-Girardeau)

---

[7]A full treatment going beyond the hydrodynamic approximation has been performed for one-body losses in homogeneous 1D quasi-condensates [4, 5].

[8]Note however that the temperature used in the definition of $D$ refers to the phononic modes only.

regime. Here, ones expects that the losses act in a similar way as in a non-interacting Fermi gas. One-body losses, for example, should then produce heating, since the temperature increases as the degeneracy of an ideal Fermi gas decreases. Finally, it would be interesting to investigate whether the results presented here may also cover interacting Fermi gases in the superfluid regime.

# Acknowledgements

M. S. gratefully acknowledges support by the *Studienstiftung des deutschen Volkes*. This work was supported by Région Île de France (DIM NanoK, Atocirc project). The work of C. H. is supported by the *Deutsche Forschungsgemeinschaft* (grant nos. Schm 1049/7-1 and Fo 703/2-1).

# A    Reduction to low-dimensional hydrodynamics

As mentioned in the main text, we assume the loss process is slow enough so that, first, the mean profile at each time is very close to the equilibrium profile with the same atom number, and second, we can safely neglect any mean velocity field when computing the time evolution of the fluctuating fields $\delta n$, $\varphi$. The evolution equations $\partial \delta n/\partial t$ and $\partial \varphi/\partial t$ are thus, at a given time, equal to those for a time-independent quasi-condensate. In the purely 3D, 2D and 1D cases, for contact interactions, we can use the well known results based on Bogoliubov theory. We then find that the equation of state takes the form $\mu = gn$ and $\partial \delta n/\partial t$ and $\partial \varphi/\partial t$ derive from Eq. (12) for the long-wavelength modes.

Let us now consider the case where the gas is confined strongly enough in 1 or 2 dimensions, such that the relevant low-lying excitations are of planar or axial nature. We allow, however, for a transverse broadening of the wave function under the effect of interactions. We show below that the equations of motion for the slow phononic modes, for which the transverse shape adiabatically follows the density oscillations, also derive from Eq. (12). The proof given here is complementary to Refs. [10,11] because it does not need an explicit model about the shape of the transverse wave function. In order to simplify the notations, we restrict ourselves to the quasi-1D situation. The derivation can be easily translated to quasi-2D situations.

We thus consider a gas confined in a separable potential consisting of a strong transverse confinement and a smooth longitudinal confinement. The equilibrium density distribution of the quasi-condensate is $|\phi_0(x,y,z)|^2$ where the real function $\phi_0(x,y,z)$ obeys the stationary Gross-Pitaevskii equation

$$\left(-\frac{\hbar^2}{2m}\partial_z^2 - \frac{\hbar^2}{2m}\Delta_\perp + V_\perp(x,y) + V(z) + g|\phi_0|^2 - \mu_p\right)\phi_0 = 0. \tag{38}$$

Here $g = 4\pi\hbar^2 a/m$ is the 3D coupling constant with $a$ the zero-energy scattering length. Within the Bogoliubov theory, the evolution of excitations is governed by the equations [19]

$$\begin{cases} i\hbar\partial_t \tilde{f}^+ = \left(-\frac{\hbar^2}{2m}\partial_z^2 - \frac{\hbar^2}{2m}\Delta_\perp + V_\perp(x,y) + V(z) + g|\phi_0|^2 - \mu_p\right)\tilde{f}^- \\ i\hbar\partial_t \tilde{f}^- = \left(-\frac{\hbar^2}{2m}\partial_z^2 - \frac{\hbar^2}{2m}\Delta_\perp + V_\perp(x,y) + V(z) + 3g|\phi_0|^2 - \mu_p\right)\tilde{f}^+. \end{cases} \tag{39}$$

The field operators are half sum and difference of the fluctuating field operators $\delta\psi$ and $\delta\psi^\dagger$. $\tilde{f}^+$ is linked to density fluctuations and $\tilde{f}^-$ to phase fluctuations.

Since we assume that the axial variation is slow compared to the transverse one, the solution $\phi_0$ can be approximated by a function $\psi$ that depends on the axial coordinate $z$ only via a local chemical potential

$$\phi_0(x,y,z) \simeq \psi(x,y;\mu), \qquad \mu = \mu_p - V(z). \tag{40}$$

Here, $\psi$ solves the Gross-Pitaevskii equation for an axially homogeneous system:

$$\left(-\frac{\hbar^2}{2m}\Delta_\perp + V_\perp(x,y) + g|\psi|^2 - \mu\right)\psi = 0. \tag{41}$$

This procedure is consistent, e.g., with making the Thomas-Fermi approximation in the axial direction. Solving this equation yields the local chemical potential as a function of the axial (average) density $\mu = \mu(n_0)$ with

$$n_0(z) = \int dx dy \, |\phi_0(x,y,z)|^2 \simeq \int dx dy \, |\psi(x,y;\mu)|^2. \tag{42}$$

This motivates the following separation *Ansatz* for the Bogoliubov functions in Eq.(39):

$$\begin{cases} \tilde{f}^+ = \partial_\mu \psi \partial_n \mu F^+ \\ \tilde{f}^- = i\phi_0 F^-. \end{cases} \tag{43}$$

where the functions $F^+$ and $F^-$ depend only on $z$ and the derivative $\partial_n \mu$ is evaluated at the local density $n_0$. Inserting this into the second line of Eq.(39), we find

$$-\phi_0 \hbar \partial_t F^- = \left(-\frac{\hbar^2}{2m}\partial_z^2 - \frac{\hbar^2}{2m}\Delta_\perp + V_\perp(x,y) + V(z) + 3g|\phi_0|^2 - \mu_p\right)\left(\partial_\mu \psi \partial_n \mu F^+\right). \tag{44}$$

The action of this operator on $\partial_\mu \psi$ can be worked out by differentiating Eq. (41) versus $\mu$: this gives

$$\left(-\frac{\hbar^2}{2m}\Delta_\perp + V_\perp(x,y) + 3g|\psi|^2 - \mu\right)\partial_\mu \psi = \psi \simeq \phi_0. \tag{45}$$

Eq.(44) thus simplifies into

$$-\phi_0 \hbar \partial_t F^- = -\frac{\hbar^2}{2m}\partial_z^2\left(\partial_\mu \psi \partial_n \mu F^+\right) + \phi_0 \partial_n \mu F^+. \tag{46}$$

To find a closed equation for the axial dynamics, we multiply with $\psi(x,y;\mu)$ and integrate over the transverse coordinates. Using Eq.(42) and its derivatives with respect to $\mu$ and $z$, we find the identities

$$\int dx dy \, \phi_0 \partial_\mu \psi = \frac{1}{2}\partial_\mu n_0 = \frac{1}{2\partial_n \mu}, \qquad \int dx dy \, \phi_0 \partial_z \phi_0 = \frac{1}{2}\partial_z n_0. \tag{47}$$

Using the first one, Eq.(46) becomes:

$$-\hbar \partial_t F^- = -\frac{\hbar^2}{4mn_0}\partial_z^2 F^+ + \partial_n \mu F^+ \simeq -\partial_n \mu F^+, \tag{48}$$

where in the second step, we took the long-wavelength limit.

Let us now insert the *Ansatz* (43) into the first line of Eq.(39):

$$\hbar \partial_\mu \psi \partial_n \mu \partial_t F^+ = \left(-\frac{\hbar^2}{2m}\partial_z^2 - \frac{\hbar^2}{2m}\Delta_\perp + V_\perp(x,y) + V(z) + g|\phi_0|^2 - \mu_p\right)\left(\phi_0 F^-\right). \tag{49}$$

The action of the operator in parentheses on $\phi_0$ simply vanishes because this is the Gross-Pitaevskii equation (38). Since $F^-$ does only depend on the axial coordinate, we are left with:

$$\partial_\mu \psi \partial_n \mu \partial_t F^+ = -\frac{\hbar}{m}\left(\partial_z \phi_0\right)\partial_z F^- - \frac{\hbar}{2m}\phi_0 \partial_z^2 F^-. \tag{50}$$

We again project out the transverse coordinates and use the identities (47). Combining the axial derivatives, we then have

$$\partial_t F^+ = -\frac{\hbar}{m}\partial_z(n_0 \partial_z F^-). \tag{51}$$

These calculations illustrate that the *Ansatz* of Eq.(43) captures well the axial and transverse dependence of the collective excitations in the low-dimensional gas. Note in particular how the density fluctuations ($\tilde{f}^+$) are accompanied by density-dependent changes in the transverse wave function.

To make contact with the hydrodynamic Hamiltonian (12), we need to relate $F^+$ and $F^-$ to the low-dimensional density and phase fields, $\delta n$ and $\varphi$. Bogoliubov theory tells us that three-dimensional density fluctuations are linked to $\tilde{f}^+$ via $\delta\rho = 2\phi_0\tilde{f}^+$. Integrating $\delta\rho$ over the transverse plane, replacing $\tilde{f}^+$ by its Ansatz (43) and using Eq. (47), we obtain

$$F^+ = \delta n = n - n_0. \tag{52}$$

Phase fluctuations on the other hand are linked to $\tilde{f}^-$ according to $\tilde{f}^- = i\phi_0\varphi$. [Recall that the ansatz (43) assumes a uniform phase in the $x, y$ plane.] Comparison with Eq. (43) gives immediately

$$F^- = \varphi. \tag{53}$$

Then Eq.(48) and Eq.(51) are precisely the evolution equations derived from the Hamiltonian (12).

## B Hydrodynamic Bogoliubov modes

Here we consider low-energy modes of either a three-dimensional gas or low-dimensional gas, whose dynamics is described by the hydrodynamic approximation. More precisely, we diagonalize the Hamiltonian (12), for a given, time-independent, equilibrium profile $n_0(\mathbf{r})$. From Eq.(12) we derive the evolution equations

$$\frac{\partial}{\partial t}\begin{pmatrix} \delta n/\sqrt{n_0} \\ \sqrt{n_0}\varphi \end{pmatrix} = \mathscr{L}\begin{pmatrix} \delta n/\sqrt{n_0} \\ \sqrt{n_0}\varphi \end{pmatrix}, \tag{54}$$

where

$$\mathscr{L} = \begin{pmatrix} 0 & -\frac{\hbar}{m\sqrt{n_0}}\nabla\cdot\left(n_0\nabla\left(\frac{1}{\sqrt{n_0}}\cdot\right)\right) \\ -mc^2/\hbar & 0 \end{pmatrix}. \tag{55}$$

The factors $\sqrt{n_0}$ are convenient to give the two components the same dimension and to symmetrize the differential operator that appears in $\mathscr{L}$. The two equations derived from Eq.(54) correspond to the hydrodynamic equations provided we identify $\hbar\nabla\varphi/m$ with the velocity: the first one is the continuity equation, the second one gives the Euler equation.

We build the mode expansion on pairs of real functions that form right eigenvectors of $\mathscr{L}$:

$$\mathscr{L}\begin{pmatrix} f_\nu^+ \\ if_\nu^- \end{pmatrix} = i\omega_\nu \begin{pmatrix} f_\nu^+ \\ if_\nu^- \end{pmatrix}. \tag{56}$$

Due to symmetry properties of $\mathcal{L}$, Eq.(56) entails the following properties: (a) $(f_\nu^+, -if_\nu^-)$ is a right eigenvector of $\mathcal{L}$ of eigenvalue $-i\omega_\nu$; (b) $(if_\nu^-, f_\nu^+)$ is a left eigenvector of same eigenvalue; and (c) different right eigenvectors of $\mathcal{L}$ verify $\int d^d\mathbf{r}\, f_\nu^- f_{\nu'}^+ = 0$. It is convenient to consider those eigenvectors of $\mathcal{L}$ which are normalized according to $\int d^d\mathbf{r}\, f_\nu^- f_\nu^+ = 1$. This yields the expansions

$$
\begin{pmatrix} \delta n/\sqrt{n_0} \\ \sqrt{n_0}\,\varphi \end{pmatrix} = \frac{1}{\sqrt{2}} \sum_\nu \left\{ a_\nu \begin{pmatrix} f_\nu^+ \\ -if_\nu^- \end{pmatrix} + a_\nu^+ \begin{pmatrix} f_\nu^+ \\ if_\nu^- \end{pmatrix} \right\},
\tag{57}
$$

which invert into

$$
a_\nu = \frac{1}{\sqrt{2}} \int d^d\mathbf{r} \left( \frac{\delta n(\mathbf{r})}{\sqrt{n_0}} f_\nu^-(\mathbf{r}) + i\sqrt{n_0}\,\varphi(\mathbf{r}) f_\nu^+(\mathbf{r}) \right).
\tag{58}
$$

The normalisation of the eigenvectors and the relation $[\delta n(z), \varphi(z')] = i\delta(z - z')$ ensure $[a_{\nu'}, a_\nu^\dagger] = \delta_{\nu',\nu}$.

We introduce the function

$$
g_\nu = \sqrt{n_0}\, f_\nu^+,
\tag{59}
$$

and use the relation $f_\nu^- = mc^2 f_\nu^+/(\hbar\omega_\nu)$ that follows from the eigenvalue problem (56). Then the normalisation of $g_\nu$ [Eq.(14)] follows from that of $(f_\nu^+, if_\nu^-)$. Defining the quadratures $x_\nu = (a_\nu + a_\nu^\dagger)/\sqrt{2}$ and $p_\nu = -i(a_\nu - a_\nu^\dagger)/\sqrt{2}$, the expansions (57) give Eqs.(16) of the main text.

## C  Numerical calculation

For the numerical results shown in Fig.3, we have solved the Gross-Pitaevskii equation in a 1D harmonic trap by minimising the corresponding energy functional: this gives a smooth density profile $n_0(z)$. The Bogoliubov equations are solved with a finite-difference scheme on a non-uniform grid. We get a frequency spectrum that coincides to better than one percent with the Legendre spectrum for all modes with $\hbar\omega_\nu \lesssim 0.1\, gn_p$ ($n_p$ is the peak density). The traditional Bogoliubov modes $u_\nu$ and $v_\nu$ are related to the eigenfunctions of Eq.(56) by

$$
\begin{aligned}
f_\nu^+ &= \sqrt{2}(u_\nu + v_\nu), \tag{60} \\
f_\nu^- &= (u_\nu - v_\nu)/\sqrt{2}. \tag{61}
\end{aligned}
$$

Inserting this into Eq.(59) gives the modes $g_\nu$. We have checked for phonon excitations with frequencies $\hbar\omega_\nu \ll gn_p$, that the proportionality between $f^+$ and $f^-$ [see after Eq.(59)] is an excellent approximation in the bulk of the condensate.

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
