# Peer review of "Cooling phonon modes of a Bose condensate with uniform few body losses"

_SciPost Physics, doi:SciPost Phys. 5, 043 (2018)_

## Round 2 · Referee Report · Anonymous · 2018-7-26

Strengths

1- Significant extension and generalization of previous theoretical models, which enables the interpretation of recent experiments studying cooling through losses in cold atomic gases.
2- Comprehensive, timely and relevant.
3- Clear explanation of all steps of the derivation.

Weaknesses

None (see below for minor questions)

Report

The authors present a general formalism for the cooling of quantum gases through uniform particle loss. While often associated with heating it is shown that under certain conditions few-particle loss processes can also lead to cooling. The study is motivated by a series of recent experiments, e.g. in the groups of Schmiedmayer and Bouchoule. As such, it is timely and relevant.

The present work is particularly interesting because the existing formalism to describe such cooling is significantly extended to cover general N-body losses, as well as arbitrary dimensions and realistic trapping potentials. For example, the cooling of a 1D gas through 3-body losses has recently been studied in an experiment by some of the authors for the first time. The results show good agreement with the theoretical predictions outlined in the present study. It will be very interesting to check experimentally if the predictions also hold in 3D situations, where evaporation (i.e. cooling through selective removal of particles with above average energy) will compete with the mechanism studied here (uniform loss of particles).

The study is well written and clearly explains all assumptions and steps of the derivation, as well as some limits of the theory. In particular, it nicely describes the inherent competition between the cooling of the low energy modes and their heating through shot noise.

Requested changes

A few minor questions:
1- Footnote 1: It is not quite clear to me what exactly the remark on Ito / Stratonovich formalisms refers to here? Could the authors briefly elaborate to make this clearer for the general reader?
2- Is there a simple way to estimate for which experimental parameters one would expect the assumption of Poissonian, uncorrelated shot noise to hold for the loss process?
3- In realistic experiments there will often be different loss processes present at once (e.g. three-body loss from interatomic collisions due high atomic densities, and one-body loss from background gas collisions), each with their own asymptotic temperature. What would one expect to see in these cases?
4- Is there a deeper meaning / interpretation behind the asymptotic temperature? Particularly Fig. 1b seems to suggest that the cooling exhibits some universal properties?
5- It would be nice to discuss if the gases become more degenerate during the cooling process. Can new regimes be reached, e.g. for 1D Bose gases?

  • validity: top
  • significance: high
  • originality: high
  • clarity: high
  • formatting: perfect
  • grammar: perfect

Author:  Isabelle Bouchoule  on 2018-09-10  [id 316]

(in reply to Report 1 on 2018-07-26)

We thank the referee for her/his careful reading of the manuscript and we are glad to hear that she/he appreciates the presented work. We answer below to her/his comments and questions:

  1. In the stochastic equation for $N$ (Eq. 1), the variance of the random variable $d\xi$ depends on $N$ (see Eq. 2). For a differential equation for $N$ of the form $dN = -A(N)dt + d\xi$ with an $N$-dependent stochastic term (namely $\langle d\xi(t)d\xi(t') \rangle = f(N) \delta(t-t')$), one should specify whether the stochastic equation is written in the Stratonovich or in the Ito formalism. However, since N(t) presents only small fluctuations around its mean value $N_0(t)$, one can replace $N$ by $N_0(t)$ when computing the stochastic term. Then, the resulting equation for $\delta N = N - N_0$ has a stochastic term which does not depend on $\delta N$ (see Eq. 3). In such a case the Ito and Stratonovich formalisms are equivalent.

In order not to confuse the reader not familiar with stochastic differential equations, we prefer to remove the footnote. The reader aware of the subtle details of stochastic equations will figure out himself that differences between Ito and Stratonovich equations are irrelevant here.

  1. The loss process may be assumed to be of Poissonian nature, as long as we consider it on a time interval dt small enough so that the number of lost atoms is very small compared to $N_0$: then the loss rate is constant during the time interval dt, which amounts to a Poissonian process.

  2. The question of the presence of multiple loss processes could a priori be addressed within our formalism, provided Eq.20 is modified to account for each loss process. Then Eq.23 would be modified as well. We foresee that the temperature will still take values close to $mc^2$. We believe that such a question, while interesting and relevant, deserves a separate study, and we prefer to skip it in the present paper.

  3. Unfortunately, we do not have a simple argument that shows that the ratio $k_B T/(mc^2)$ goes towards a stationary value as a result of losses. In the introduction and later in the text (after Eq.3 and after Eqs.23-26), we emphasize the underlying physics: cooling due to the decrease of density fluctuations on the one hand, and heating due to the stochastic nature of losses, on the other. To our knowledge however, only quantitative calculations permit to conclude that the relevant quantity, which takes a stationary value, is the ratio $k_B T/(mc^2)$.

  4. In the conclusion, we added a brief discussion what may eventually happen with the ultracold gas as the density and temperature are lowered. It indeed crucially depends on the system dimensionnality whether quantum degeneracy increases or decreases.

---

## Round 3 · Referee Report · Anonymous (Referee 1) · 2018-9-26

Report

The authors have satisfactorily addressed my previous comments.

---

## Round 3 · Referee Report · Anonymous (Referee 2) · 2018-10-22

Strengths

  1. Timely recent topic of significant practical and theoretical interest to the community.

  2. Significantly extends the regimes in which the cooling-by-losses phenomenon has been described to include largely arbitrary potentials and several different loss mechanisms.

  3. The main equations derived appear to be readily usable for a wide range of future problems.

  4. Very clearly written, easily understandable paper.

  5. Good segregation of what went into the main body of the text and the appendices.

Weaknesses

  1. Based on three quite strong approximations: local density, close-to-adiabatic, and phonon modes only. This will makes the results inapplicable to a significant proportion of the interesting cases.

  2. Although the equations are developed for arbitrary dimension, only 1d examples are provided.

  3. No explicit mention of the use of these equations already in Ref. [9].

Report

The paper concerns a topic that has drawn significant interest in the last two years - a cooling that appears due to particle losses in ultra-cold gases. It was noticed somewhat unexpectedly in some experiments in 2016, and has been discussed in the community since then. A cute thing about this mechanism is that provides a route to cooling below the level of the chemical potential that occurs naturally without complicated manipulations of the gas and seems that it could be made almost automatic. Also, it is appealing that such a process was not really suspected in the community beforehand and has now been spotted. There have been both some experimental and theoretical studies of it since the initial report, including an experiment by the present authors, Ref. [9].

This paper significantly extends the regimes in which the process has been described to include considerations of very arbitrary potentials, and several-particle loss. This will be helpful in enabling future researchers in making straightforward initial calculations of the phenomenon in a wide variety of systems.

The study bases itself very strongly on the local density and close-to-adiabatic approximations, which leads fairly directly to both the main strengths and main weaknesses of the results. The strength being that the resulting equations (23-26) can be applied to practically any potential and a variety of loss processes. Furthermore, these equations are not numerically problematic or onerous. The weakness is that the results thus obtained are restricted to slowly evolving phonon modes, with small amplitudes, in slowly varying potentials. In terms of space-dependence, anything with length-scales approaching healing length will not be treated well. Some of the earlier theoretical works on the subject go beyond these restrictions, but with more idealised potentials, losses. Overall the results of this work complement previous studies well, and significantly move the knowledge about the topic forward.

The equations that constitute the main result of the paper are derived as applicable to 1,2,3 dimensions, so it is a little disappointing (though not atypical) that only 1d examples have been provided. Still, it is nice to see that the authors note earnestly in the conclusions that the validity of the equations in 2d and 3d is likely to be more restricted.

Knowing that the present authors are also authors of the experimental work with cooling by three-body losses [9], for which the methods developed here are ideal, I was surprised to find no explicit calculations comparing to that experiment here. However, it turns out that Ref. [9] does cite the present manuscript extensively, referring to it actually as the source of the calculations made there. There is no explicit statement of the same fact here in the present manuscript though, leaving readers a bit in the dark. It would be very helpful for clarity to also explicitly explain also *here* that these methods have already been applied in [9].

Apart from this one little thing, the paper is very clearly written, with the right balance of explanation where necessary and abstention from writing out gruelling details where they can readily be filled in by those interested. The choice of which details are relegated into appendices worked well. Other than more explicitly explaining the relationship to Ref. [9], I don't really have any other changes to recommend. The paper is a very worthwhile standalone addition to the current knowledge on the topic as is.

Requested changes

  1. Please clarify that the methods presented here have already been applied in the experimental paper Ref. [9], and make a brief comment on how good the match was.

---

## Round 3 · Author Response

Dear editor,

Please find here a new version of our paper, with minor modifications.

Best regards,

The authors.

---

## Round 3 · List of Changes

p3: the footnote on Ito vs. Stratonovich has been suppressed.

Eq.(17), first line: for consistency with the second line, we have added
the argument r to the fields \delta n and g_\nu. Same cosmetic change in
Eqs.(24,25), Eq.(37), and Eq.(58).

p12: we have added a paragraph on the thermodynamic phases that one may expect
for the Bose gas at long times.

p13: a reference to a paper by Cockburn & al,
"Comparison between microscopic methods for finite-temperature Bose gases",
Phys. Rev. A 83, 043619 (2011)
has been suppressed as it was not relevant for the 'two-temperature' case
in question. We kept the reference [5] to Johnson & al,
"Long-lived non-thermal states realized by atom losses in one-dimensional quasi-condensates",
Phys. Rev. A 96, 013623 (2017).

---

## Round 4 · Author Response

Dear editor,

Please find here a new version of our paper that we re-submit to Scipost.

Best regards,
The authors.

---

## Round 4 · List of Changes

According to the second referee comments, we put more emphasis on the fact that results of this paper have been successfully tested in a recent experimental paper of M. Schemmer and I. Bouchoule, where the effect of 3-body losses on a 1D Bose gas is investigated (ref [9] of the manuscript). More precisely, we added a sentence at the end of the introduction and a sentence after Eq. 23-27.

---

## Editorial Decision

published